# Comparative Study of the Efficiency of Additives in the Extraction of Phycocyanin-C from *Arthrospira maxima* Using Ultrasonication

**DOI:** 10.3390/molecules28010334

**Published:** 2022-12-31

**Authors:** Jorge Eugenio Martínez-Vega, Erika Villafaña-Estarrón, Froylán M. E. Escalante

**Affiliations:** 1Departamento de Biotecnológicas y Ambientales, Universidad Autónoma de Guadalajara, Av. Patria 1201, Lomas del Valle, Zapopan 45129, Mexico; 2Dirección de Investigación y Desarrollo Tecnológico, Universidad Autónoma de Guadalajara, Av. Patria 1201, Lomas del Valle, Zapopan 45129, Mexico

**Keywords:** ultrasonication, freeze–thaw, *Spirulina*

## Abstract

Several phycocyanin extraction methods have been proposed, however, most of them present economical or productive barriers. One of the most promising methods that has been suggested is ultrasonication. We have analyzed here the effect of operational conditions and additives on the extraction and purity of phycocyanin from *Arthrospira maxima*. We followed three experimental designs to determine the best combination of buffered pH solutions, additives, fresh and lyophilized biomass. We have found that additives such as citric acid and/or disaccharides could be beneficial to the extraction process. We concluded that the biomass–solvent ratio is a determining factor to obtain high extraction and purity ratios with short ultrasonication times.

## 1. Introduction

Pigments are high-value molecules in the industrial sector, including in food, cosmetics, and pharmaceutics, and are widely used mostly to enhance product appearance. Plants and microorganisms are the two major sources of natural pigments. Among the most promising microorganisms for their production are microalgae; some of the advantages of these organisms as pigment producers are the ease of growth and cultivation in bioreactors, as well as the capacity to produce a wide range of natural and safe colorants [1].

In this regard, phycobiliproteins are a special kind of natural pigment with nutraceutical activity and pharmaceutical applications. These molecules achieved a market value of 18.5 million USD in 2018 with an expected growth rate of 8.6% given their potential use as beverage and cosmetics additives [2]. Because of this, and its varied industrial applications, phycocyanin-C has gained a lot of interest in its obtention and stabilization.

Phycocyanin-C is generally extracted from cyanobacteria such as (*Spirulina*) *Arthrospira* sp.; other alternatives are cyanobacteria *Synechococcus* sp. PCC 6715 (*Thermostichus lividus* PCC 6715) or the microalgae *Galderia sulphuraria* and *Cyanidioschyzon merolae*, these last with outstanding properties given their tolerance to extreme environments [3,4,5]. Traditionally, this pigment is easily extracted in aqueous solutions given its solubility in water; the freeze–thawing method is the easiest, and thus the most commonly used [6]. However, this method presents obstacles in keeping the stability of the pigment, mainly because of the environmental lability of the phycocyanin. When the molecule is exposed to light, heat, or pH changes it loses its color, and thus its nutraceutical properties [7,8]. Considering this, new alternatives have been proposed to extract it from the cell matrix such as enzymatic digestion, high-pressure processing, ion-exchange chromatography, mortar grinding, pulsed electric fields, ultrasonication, and ultrafiltration [5,6,9,10]. It has been suggested, that ultrasonication extracts more intact phycocyanin-C in comparison to water extraction [11]. In most cases, the addition of preservatives (ascorbic acid, citric acid, calcium chloride, sucrose, sorbitol, and trehalose) or buffer solutions (phosphate or acetate) has been advised [3,9,12,13]. Other authors have gone further and suggested that there is an effect of using wet or dry biomass [14]. The constant in all these methods seems to be the use of buffers to keep the pH of the extraction solution around 6–7.

In this report, we present a comparative study carried out to evaluate several factors which might impact the extraction yield and purity of phycocyanin-C from *Arthrospira maxima*. There are multiple studies on additives or new methods of extraction to increase the extractive efficiency and/or avoid the degradation of the pigment. Nonetheless, here a series of multifactorial and multiple response optimization statistical analyses is presented, leading us to study the simultaneous effect of several factors on the extraction efficiency and purity of the pigment. The main objective was to determine the best processing treatment for the biomass of *A. maxima* based on the most recurrent parameters reported in the literature.

## 2. Results

### 2.1. Screening of Factors with Impact on Phycocyanin Extraction Yield Using Sonication as a Cell-Disruption Method

It is well known from previous reports that buffer solutions and additives are required for the extraction of phycocyanin. In this first set of experiments we started analyzing two pH buffers and three of the most common additives to extract phycocyanin-C from lyophilized biomass in a solution-volume/biomass rate of 20:1 v/w.

The highest yield of phycocyanin/biomass (1.65%) was observed using a mixture of citric acid and trehalose in the extraction solution, Figure 1B. It can be seen also that buffered solutions at pH 5 were more effective in terms of extraction yields in comparison to those at pH 7. In general, additives in the lowest concentration led to higher extraction rates. All the results were tested against a statistical significance of 95% (*p* < 0.05).

Since all the interactions among the three factors were statistically significant (*p* < 0.05), each interaction was plotted (Figure 2); from them, it can be concluded that higher extraction yields can be obtained after using buffered solutions at pH 5 added with a mixture 0.1% of citric acid + trehalose. The addition of sucrose or trehalose alone did not show significant differences between the tested levels, and their extraction yields were the lowest.

A second set of experiments was carried out using a solution-volume/biomass ratio of 200:1 v/w. Considering the previous result, where the lower pH gave better extraction yields, we next tested buffered solutions at pH 4 and 5 to respect the tendency of better extractions using lower pH values. Again, the mixture of citric acid + trehalose and trehalose alone were used as additives in the same concentrations as before (0.1 and 0.5%). We could not detect a significative difference (*p* < 0.05) of the additives’ concentration; however, when trehalose was used alone, higher extraction yields were obtained, 3.98 ± 0.06%. In general, the extraction rates were three times those observed in the previous experiment, as shown in Figure 3.

### 2.2. Analysis of the Effect of Sonication Parameters in Phycocyanin Extraction Yield and Purity

After the previous experiments, we decided to analyze the effect of sonication cycles (2 and 4) and the time of each cycle (30 and 60 s). In this case, since pH 4 yielded poor results, we analyzed again pH in the range of 5 to 7, and trehalose in the previous levels (0.15 and 0.3%). This time we observed that pH 7 was the most efficient value, both for extraction yield and purity. This time, the results were not different in response to the changes of trehalose concentration, either in extraction efficiency or purity of the pigment. Nonetheless, we could notice that sonication cycles and time of treatment had a positive impact on the pigment’s extraction (Figure 4A), but not on the purity (Figure 4B).

Since all two-factor combinations were significant (*p* < 0.05) for extraction yield, we determined the best combination to be four sonication cycles of 30 s, or two cycles of 60 s, e.g., two minutes cellular disruption treatment (Figure 5A–C). A similar effect was observed for purity, where the best option was to use the largest sonication times (Figure 5D).

After applying the response surface methodology, we could determine the best combination of the factor levels to be pH = 7, trehalose concentration = 0.3, sonication cycles = 4, and sonication time = 60 s for each cycle. With this it is possible to achieve an extraction yield of 5.7% phycocyanin/biomass. It must be considered that according to the optimized model, as long as pH is kept near 7, the remaining factors could be altered as convenient (Figure 6). A similar result was obtained for phycocyanin purity with a maximum value of 0.63.

### 2.3. Analysis of the Phycocyanin Extraction Yield and Purity when Combining Sonication with Other Operation Parameters

In this set of experiments, lyophilized or wet biomass was suspended in pH 7 buffered solutions. Given the rise in temperature of the extraction solutions during the sonication process, we analyzed the effect of submerging the sample tubes into an ice-water bath at this point of the extraction. Additionally, we tested the effect of a post-extraction procedure using the freeze–thaw method. As shown in Figure 7A, the use of an ice-water bath and non-lyophilized biomass (wet biomass) resulted in higher extraction yields. On the contrary, the freeze–thaw procedure did not increase the concentration of phycocyanin in the solution by itself. Conversely, the freeze–thaw procedure is advisable, particularly when working with lyophilized biomass, Figure 7B.

Also, we analyzed the purity of the extracted phycocyanin in the abovementioned experiment. In this case, the initial state of the biomass (wet or lyophilized), as well as the ice bath, had significative impacts on this parameter. We could measure purity of 0.88 ± 0.05 with treatments of the ice-water bath or wet biomass (non-lyophilized), as shown in Figure 8A. It is interesting to notice that an ice-water bath is necessary to obtain a good level of purity (0.88) when using lyophilized biomass, in contrast with wet biomass where the ice-water bath seems to be unnecessary or has no impact (Figure 8B).

## 3. Discussion

### 3.1. Screening of Factors with Impact on Phycocyanin Extraction Yield Using Sonication as a Cell-Disruption Method

In this first set of experiments we studied the effect of two buffered solutions, and three additives: citric acid, sucrose, and trehalose. Adjali et al. [7] summarized several reports where pH levels ranged mostly between 5 and 7; in this work, we decided to test the effect of buffered solutions at these two levels. Our results for this first set of experiments showed that pH 5 was the best choice; this was in accordance with Wu et al. [15] who reported an optimal pH range from 5 to 6 for the extraction of the pigment from *Spirulina platensis*. Kumar et al. [16] also reported the extraction in acetate buffer at pH 5.10 to be the best choice; in contrast, Li et al. [9] reported a better extraction yield at pH 7.5. The results are someway contradictory/controversial, since in the following set of experiments presented in this report, we found a better extraction at pH 7. In general, given the higher solubility of phycocyanin at pH 7, it has been suggested as the optimal value [17,18,19]. Then, we hypothesize that even when pH is one of the most important factors on any extraction process of phycocyanin, the extraction conditions impact on the efficiency and set point of this parameter, i.e., buffer concentration, solvent–biomass ratio, temperature, etc. [18]. pH values above 7 are not recommended since the phycocyanin degrades quickly. Higher pH values (7.5 and 8) were studied before [9], and the authors report that even when pHs above 7 are faster in extracting phycocyanin, the pigment is unstable and suggest to adjust pH of the final solution to about 6.0 and 6.5 to preserve the pigment.

In the same experiment we analyzed the combined effect of three additives with pH. Even though additives are widely considered as long-term stabilizers, we decided to include citric acid, sucrose and trehalose hypothesizing that these could improve the stability of the pigment during the extraction procedures, given that sonication increases the temperature of the solutions, and because it has been suggested that phycocyanin is less thermostable at pH 7 [4]. Despite the lack of information related to the effect of stabilizers in the extraction of the phycocyanin, we could observe a higher extraction rate when citric acid + trehalose were added to the solution of pH 5, prior to their processing.

We repeated the experiments of the first stage using this time buffered solutions of pH 4 and 5. Again, an extraction rate around 1.8% (18 mg/g) was observed when the solution pH 5 was added with citric acid + trehalose, however, at the same pH, the use of trehalose alone increased the efficiency up to 4% (40 mg/g) despite the pH of the solution. The only other variant in this set was the biomass–solvent ratio, which was increased ten times, i.e., went from 20:1 (0.05 g/mL) to 200:1 (0.005 g/mL). This indicates that the biomass–solvent ratio is as important as the pH of the solution to improve the yield of phycocyanin-C. Similarly, a ratio of 100:1 had been previously assayed [20,21], and it was reported that an increase in the biomass–solvent ratio led to an increase in the phycocyanin concentration. The most recent works suggest 0.08 g/mL as the best ratio [18,19,22], with the largest amount of extracted phycocyanin at around 4.7% (46.8 mg/g) using a phosphate buffer solution of pH 7 and this dilution rate [22].

### 3.2. Analysis of the Effect of Sonication Parameters in Phycocyanin Extraction Yield and Purity

Since other works using sonication report processing times around 10 min and biomas–solvent ratios of 0.08 g/mL, we decided to evaluate the effect of sonication cycles and time of the cycle. We tested this in combination with pH and addition of trehalose. We found no difference between two or four cycles of 60 s, either in extraction or purity ratios. Ores et al. (2016) used an ultrasonication bath of 20 kHz (60 W of power) and glass beads for 10 min with extraction efficiency of 47 mg/g; Aftari et al. (2015) used 20 kHz (100 W) for 10 min (2.84 mg/mL, with a biomass–solvent ratio of 0.08), and also 50 kHz for 40 min was used, achieving 0.57 mg/g or 43.75 mg/g without and with glass beads [23]. In this work we used 20 kHz (130 W) for 2 min, reducing significatively the time of treatment and achieving extraction rates of 40 mg/g. Li et al. (2020) suggest that the sonication amplitude plays an important role in disrupting the cell walls of *Spirulina*. This led us to hypothesize that shorter sonication times with high amplitudes will promote higher efficiencies of extraction and/or purity when lower biomass–solvent ratios are used; this could be explained by the fact that the ultrasonication method is based on liquid-shear forces caused by the emission of high-frequency wave sounds [24], which will be more suitable in a less concentrated media.

In this work, the best result for ultrasonication treatment was obtained at pH 7 (57 mg phycocyanin/g biomass), despite the other factors (sonication time and cycles, or addition of trehalose). However, the purity ratio was sacrificed and decreased from 1.14 (first set of experiments) to 0.63. The negative effect of increased temperature and time of ultrasonication on purity was previously reported [19]. Other works report similar values, such as that of Aftari et al. (2015) with 1.27 using microwave extraction, and 0.65 using sonication [19] (the latter being similar to that obtained in this work) or Pan-Utai et al. (2018), who obtained a purity ratio of 1.01 using lyophilized biomass [18].

### 3.3. Analysis of the Phycocyanin Extraction Yield and Purity when Combining Sonication with Other Operation Parameters

Since there are several reports for the extraction of phycocyanin from freeze-dried, oven-dried, and/or fresh (wet) biomass, we decided to test the effect of the state of biomass on extraction efficiency. We have shown here that a major content of phycocyanin can be obtained from fresh biomass in comparison to freeze-dried, however, a cycle of freeze–thaw on ultrasonicated freeze-dried biomass would yield comparable results. Previously, it was suggested that the use of fresh biomass is advisable in comparison to dry biomass given the lability of phycocyanin [25]. According to [21] extractions with fresh and lyophilized biomass give similar extraction efficiencies. However, as shown here, the purity ratio diminished from 1.04 in fresh biomass to 0.77 in lyophilized biomass [21]; in this study we obtained purities of 0.9 and 0.54, respectively; nonetheless, when phycocyanin was extracted using sonication into an ice-bath, the purity ratio for lyophilized biomass was 0.88. Both raw biomass processing methods present cost-related disadvantages, i.e., fresh biomass presents high storage costs, while freeze-dry is associated with the high costs of drying and keeping a low temperature during extraction, as pointed out by Choi and Lee [11]. It has been suggested that small saccharides could maintain the native structure of proteins in aqueous solutions even at high temperatures [26]. We observed a slightly higher purity ratio when trehalose was added to the extractive solution suggesting that osmoprotectants could improve this value.

## 4. Materials and Methods

### 4.1. Cyanobacteria Growth Conditions

*A. maxima*, fresh and lyophilized, was kindly provided by Tecnología Ambiental de Microalgas SA de CV (Biomex, Mexico) and was grown in axenic cultures in Zarrouk’s medium. Inoculant cultures were grown at 24 ± 2 °C under light/dark cycles (12/12 h) at a photon flux density (PPF) of 60 µmol m^2^/s and airflow of 1 vvm. Fresh biomass was also lyophilized at –80 °C for 12 h in several cycles until completely dried.

### 4.2. Experimental Designs and Processing Conditions

For the first set of experiments to test the effect of several parameters in the extraction of phycocyanin, 500 mg of lyophilized biomass were placed into a 15 mL conical tube with 10 mL of the corresponding extraction-solution according to a multifactorial experimental design 2 × 3 × 2; this is, two buffered solutions (pH 5 and 7) with one of three possible additives at two concentrations: citric acid/trehalose (0.1, 0.5% *m/v*), sucrose (5, 25% *m/v*), and trehalose (0.1, 0.5% *m/v*). Following, each mixture was sonicated two times at 20 kHz for 1 min at an amplitude of 70% using an ultrasonic processor 130 W (Sigma-Aldrich/Merck, Darmstadt, Germany). After this, the samples were subjected to a double thaw–freeze procedure; every tube was frozen at –20 °C for 16 h, then thawed in a water bath at 30 °C.

In the second set of experiments, to evaluate the effect of sonication parameters a 2^4^ factorial design was used. Similar conditions as mentioned above were used, however, the biomass weight was 50 mg. The factors and their levels were as follows: sonication cycles (2, 4), sonication time (30, 60 s), trehalose concentration (0.15, 0.3%), pH (5, 7).

Finally, to test the effect of the operational parameters in the extraction and purity of phycocyanin subject to ultrasonication, 50 mg of fresh or lyophilized biomass was placed into a 15 mL conical tube with 10 mL of a phosphate buffer solution (pH 7) according p according to a multifactorial experimental design 2 × 2 × 2 × 2, as follows: raw material (fresh, lyophilized biomass), ice-water bath (no, yes), freeze–thaw (no, yes), trehalose 0.3% (no, yes). Sonication and freeze–thaw followed the above-described conditions.

### 4.3. Analytical Methods

All samples were centrifuged at 8000 rpm for 10 min, and the supernatant was recovered for phycocyanin analysis. The phycocyanin content (mg/mL) and purity ratio were calculated from the optical densities at 652, 620 and 280 nm using Equations (1) and (2) according to [27]:(1)$$\mathrm{PC}=\frac{\left({\mathrm{OD}}_{620}-0.474{\mathrm{OD}}_{652}\right)}{5.34}$$
(2)$$\mathrm{Purity}\;=\frac{{\mathrm{OD}}_{620}}{{\mathrm{OD}}_{280}}$$

### 4.4. Statistical Analysis

To validate the results reproducibility, each essay was performed in duplicate. The results were treated by multifactor analysis of variance (ANOVA) followed by LSD’s post-hoc test, using Statgraphics Centurion 18 (Statgraphics Technologies, Inc. 1982-2018). All analyses were performed considering a level of 95% of confidence (*p* < 0.05).

## 5. Conclusions

In this work we could found five important factors affecting the extraction efficiency of phycocyanin from *A. maxima*: biomass/solution ratio, pH, water content of the biomass, temperature of the extraction process, and addition of preservatives prior to the extraction. Biomass preprocessing and temperature of extraction must be considered as important; even when dried (or lyophilized) biomass is preferred, wet biomass gives higher yields, and the addition of antioxidants or osmoprotectants prior to the extraction procedure when using dried biomass is advised; this could help to protect the structure of the extracted phycocyanin under stressful conditions and avoid excessive costs associated with storage of the material or lowering temperatures during the extraction process. Finally, we conclude that the most important factor to increase the efficiency of sonication extraction is the biomass/solution ratio since sonication is more efficient in more diluted solutions, leading to shorter processing times, thus to lower production costs.

## Figures and Tables

**Figure 1 molecules-28-00334-f001:**
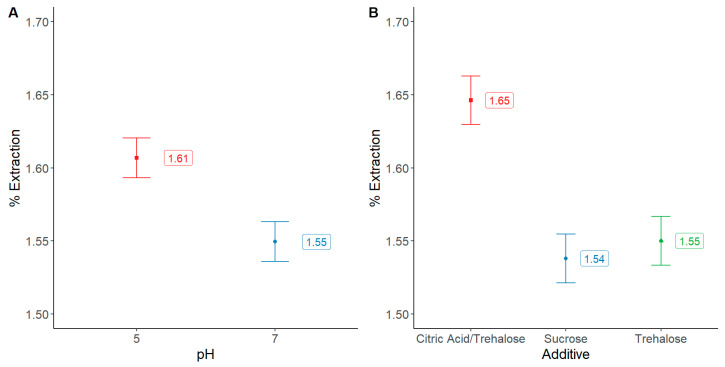
Comparative extraction yields of phycocyanin (%) as a function of (**A**) pH and (**B**) additives. Bars show means and least significant difference intervals for 95% confidence; non overlapped bars are statistically different among them.

**Figure 2 molecules-28-00334-f002:**
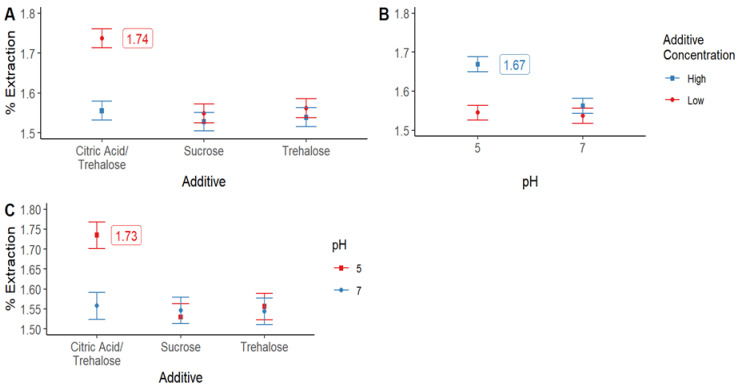
Interaction plots for (**A**) additive vs. additive concentration (trehalose and citric acid, Low = 0.1% High = 0.5%; sucrose, Low = 5% High = 25%); (**B**) pH vs. additive concentration; and (**C**) additive vs. pH. Bars show means and least significant difference intervals for 95% confidence.

**Figure 3 molecules-28-00334-f003:**
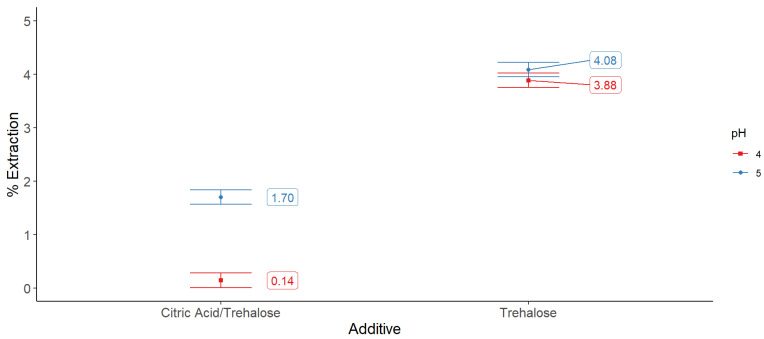
Interaction plot for additives and pH as extraction factors. All bars show the least significant difference intervals for 95% confidence.

**Figure 4 molecules-28-00334-f004:**
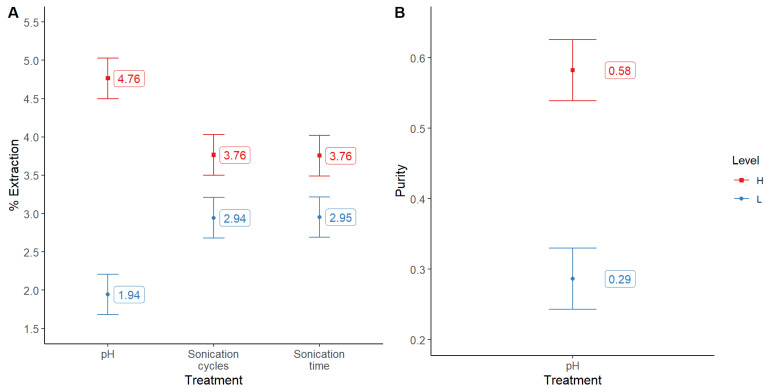
Comparative extraction yields (**A**) of phycocyanin (%) as function of several factors (pH, L = 5 H = 7; sonication cycles, L = 2 H = 4; sonication time, L = 30 s H = 60 s), and (**B**) purity of the pigment as function of pH level. Bars show the means and least significant difference intervals for 95% confidence.

**Figure 5 molecules-28-00334-f005:**
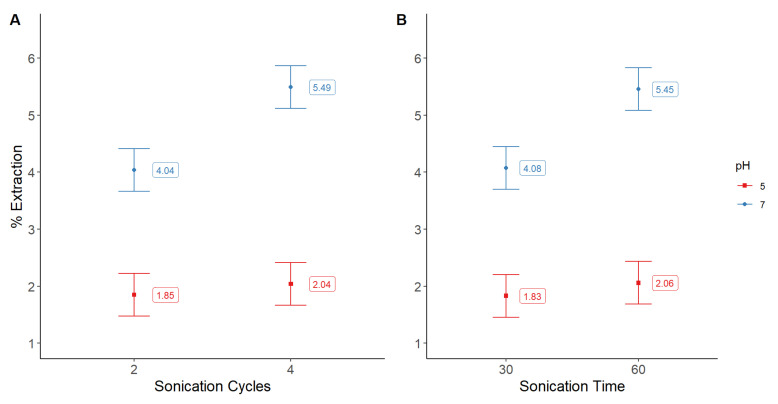
Interaction plots for yields of phycocyanin (%) (**A**) Sonication cycles vs. pH; (**B**) Sonication time vs. pH; (**C**) Sonication time vs. Sonication cycles. Interaction plot for purity of the extracted phycocyanin, (**D**) Sonication time vs. Sonication cycles. Bars show the means and least significant difference intervals for 95% confidence.

**Figure 6 molecules-28-00334-f006:**
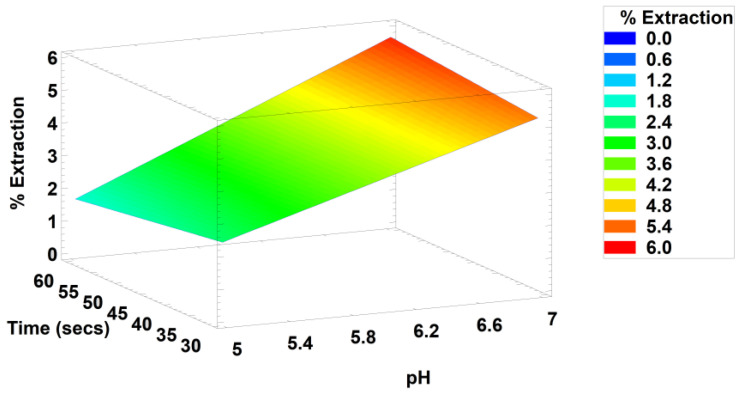
Surface response for the percentage of phycocyanin-C extracted in a buffered solution added with 0.30% trehalose subject to four sonication cycles.

**Figure 7 molecules-28-00334-f007:**
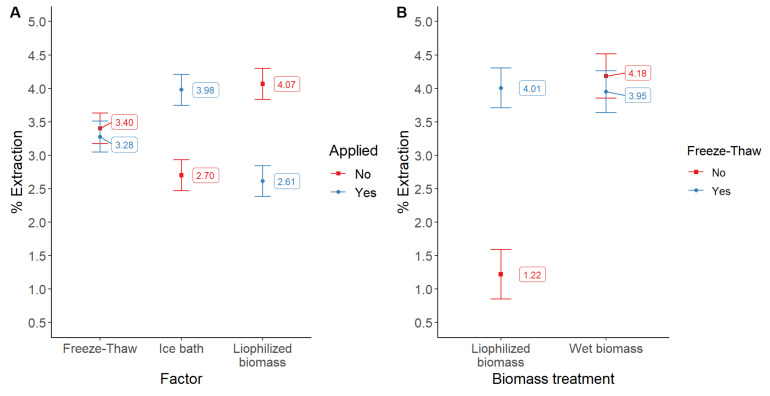
Comparative extraction yields of phycocyanin (%) as a function of three operational parameters (**A**), and (**B**) as a function of the interaction of biomass treatment vs. freeze–thaw procedure application. Bars show the least significant difference intervals for 95% confidence.

**Figure 8 molecules-28-00334-f008:**
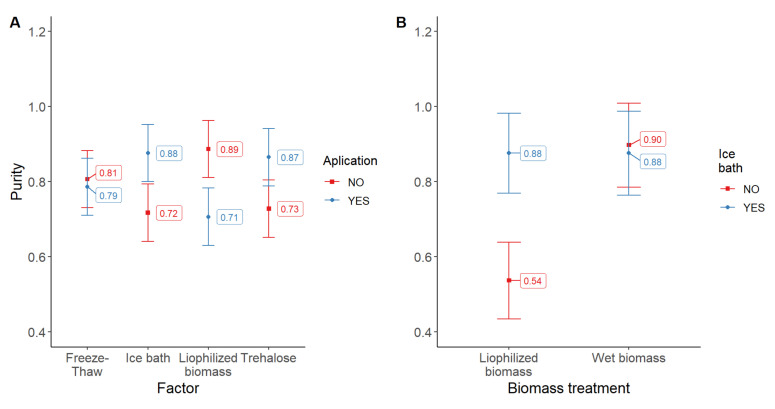
Comparative purities of phycocyanin (**A**) as a function of three operational factors and trehalose as osmoprotectant; and (**B**) as a function of the interaction between biomass treatment versus ice-water bath application. Bars show the least significant difference intervals for 95% confidence.

## Data Availability

Not applicable.

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
