# Peer review of "Comparative Study of the Efficiency of Additives in the Extraction of Phycocyanin-C from Arthrospira maxima Using Ultrasonication"

_molecules, 2022, doi:10.3390/molecules28010334_

Round 1
Reviewer 1 Report
The work entitled “Comparative study of the efficiency of additives in the extraction of phycocyanin-C from Arthospira maxima using ultrasonication” describes the extraction of phycocyanin-C using different experimental conditions such as the use of buffered pH solutions, additives and using fresh and freeze-dried biomass. The authors analyze the effect of the mentioned variables on the percentage of extraction and purity of phycocyanin-C. The work is well presented and written. The experimental part is adequate, and the correct statistical methods have been applied. There are only two aspects that the authors must correct for the work to be accepted for publication in Molecules.
1.- The caption to Figure 8 is missing.
2.- The inclusion of a section with the conclusions is also strongly recommended.
Once these two aspects are included, the work is suitable for publication.
Author Response
Dear reviewer
We appreciate your comments and we would like to let you know that we have attended both of them. Figure 8 has now a caption and we added a conclusion section. We have also made other modifications as suggested by the other reviewer. We hope this new version will result more appropriate for both of you.
Reviewer 2 Report
In their manuscript, the authors optimized the extraction of phycocyanin from Arthrospira maxima cyanobacteria. The authors studied the effect of pH, type and concentration of additives, length of sonication and the initial state of biomass on the extraction yield and purity of product. It was found that citric acid or disaccharide additives enhance the extraction process. It was concluded that biomass-solvent ratio is an important factor to achieve high extraction yield and purity ratios with short ultrasonication times. Although the experiments were designed carefully, the presentation of results has to be improved before publication in Molecules.
Comments:
1. The English of the manuscript is basically adequate (as far as I can judge as a non-native English speaker), but a thorough revision and correction of the manuscript is definitely necessary. E.g. line 64-65: "It is well known from previous reports the need for buffer solutions and additives in the extraction of phycocyanin." Suggestion: "It is well known from previous reports that buffer solutions and additives are required for the extraction of phycocyanin". Line 162: "The results are someway contradictories..." Suggestion: "The results are somewhat contradictory/controversial."
2. The results are presented in graphs, but these are not always easy to interpret. In Figure 1, the combination of many different parameters was particularly confusing. It is also not clear how to interpret these figures. According to the figure captions, the Least Significant Difference intervals are presented. As far as I know, these are for pairs of data series. In this case, which pairs of data series? Overall, the presentation of the results is confusing. I recommend that the data be presented in a tables rather than in graphs.
3. The figures are very faint. Contrast should be improved.
4. Figure 3 seems to contradict the results in Figure 2. Figure 3 has higher extraction efficiencies at pH 5 than the same conditions in Fig. 2. What is the reason for this apparent contradiction?
5. line 100: "In this case, we took back pH and trehalose levels". Took back from where? I understand that also the effect of pH 4 was studied, however, this wording is sloppy.
6. It is not clear why pH 7 was the most appropriate for ultrasonic extraction (Fig. 4), while pH 4 was the most efficient in case of Figures 1-3. This seems contradictory.
7. Fig. 7 suggests that increase of pH can further improve extraction efficiency. What is preventing the pH from rising above 7?
8. On line 228 there is a badly formatted citation.
9. There is no real conclusion other than that the authors suggest further studies. The authors should summarize their conclusions. The results obtained and measured in this study should be compared with other reported studies and it should be demonstrated that the optimized extraction method indeed gives better results than the previous ones.
Author Response
Dear reviwer
We appreciate your comments to improve our paper. Please find our response to your suggestions below:
- We attended your suggestions in this point.
- We have separated data in Figure 1, and we also added lables to each bar. We consider this will help to read the figures easily. We would like to add, that we removed some factors from the first figures since their impact on the response variable was null; we think this also increases the lecture of the figures.
- In relation to your question about LSD: effectively it is considered a pairwise test since means are compared in pairs. However, this is a test for multiple factors. The test is based on a t distribution and the selected confidence interval; then the means of each treatment or combination of them is compared with the rest of the means. This is done pair by pair until complete the computation.
- We have changed the colors in the figures. We hope this change improve the appreciation of data.
- Regarding figures 2 and 3, at the beginning of the paragraph (line 93) we reported a change in the biomass/solution ratio, this time was increased ten times (from 20:1 to 200:1); then we discuss this change in the third paragraph of section 3.1. We attribute the higher efficiency to the dilution ratio, more than to other factor.
- In relation to your comment 5 ("line 100: "In this case, we took back pH and trehalose levels"...) we have changed the redaction for a better understanding of the sentence. We hope this new phrase will be more comprehensible.
- In the case of your comment 6: "6. It is not clear why pH 7 was the most appropriate for ultrasonic extraction (Fig. 4), while pH 4 was the most efficient in case of Figures 1-3. This seems contradictory." We didn't use pH 4 in the first experiments (Figures 1 and 2), pH 4 is reported only in figure 3; this was tested because of the tendency of better efficiency when decreasing pH. We have added this note in the text of the article (lines 95-96). Besides, as stated in the same paragraph, the effect could be attributed to the presence of trehalose in the solution (lines 99-101). In the third paragraph of section 3.1 we discuss this observation.
- In comment 7 you ask why we didn't increased the pH above 7. We didn't increased pH since the protein (phycocyanin) is highly unstable at pH above 7. We add this discussion at the end of the first paragraph of section 3.1.
- In case of reference in line 228 we have made an amendment. We appreciate your observation in this point.
- We have included now the conclusions section as suggested. We consider that our major contribution is to demonstrate that some factors apart from pH are very important in the extraction of the phycocyanin, specially dilution rate. We have proven that using higher dilutions the extraction using sonication is way more efficient; as we stated also in our discussion section.
We appreciate your time to review our report and we hope have answered all your comments and suggestions properly.